# MicroRNA Sequences Modulated by Beta Cell Lipid Metabolism: Implications for Type 2 Diabetes Mellitus

**DOI:** 10.3390/biology10060534

**Published:** 2021-06-15

**Authors:** Jamie M. R. Tarlton, Steven Patterson, Annette Graham

**Affiliations:** Department of Biological and Biomedical Sciences, School of Health and Life Sciences, Glasgow Caledonian University, Glasgow G4 0BA, UK; Jamie.Tarlton@gcu.ac.uk (J.M.R.T.); Steven.Patterson@gcu.ac.uk (S.P.)

**Keywords:** microRNA, lipid, lipotoxicity, islets, beta cells

## Abstract

**Simple Summary:**

At present, more than 450 million adults worldwide are living with diabetes, with a further 370 million individuals at risk of developing this condition. Diabetes is caused by loss of production of, or sensitivity to, insulin, the hormone which controls blood sugar levels. One key factor contributing to loss of insulin output from beta cells in pancreatic islets is the damaging effects of sugars and fats in the bloodstream. This review article sought to identify the changes in expression of small pieces of RNA (microRNA) which are reported to be caused in beta cells and islets by exposure to sugars and fats. These small RNA sequences alter the expression of networks of genes which can promote, or protect, against beta cell damage, and their levels in the bloodstream have also been used as markers of diabetes. The combined effects of these microRNA sequences in beta cells were predicted, and may help to inform drug discovery strategies.

**Abstract:**

Alterations in lipid metabolism within beta cells and islets contributes to dysfunction and apoptosis of beta cells, leading to loss of insulin secretion and the onset of type 2 diabetes. Over the last decade, there has been an explosion of interest in understanding the landscape of gene expression which influences beta cell function, including the importance of small non-coding microRNA sequences in this context. This review sought to identify the microRNA sequences regulated by metabolic challenges in beta cells and islets, their targets, highlight their function and assess their possible relevance as biomarkers of disease progression in diabetic individuals. Predictive analysis was used to explore networks of genes targeted by these microRNA sequences, which may offer new therapeutic strategies to protect beta cell function and delay the onset of type 2 diabetes.

## 1. Introduction

The International Diabetes Federation (IDF) Atlas (2019) indicates that there are currently 463 million adults (29–79 y) living with diabetes, predicted to rise to 700 million by 2045, with a further ~374 million people at increased risk of developing type 2 diabetes mellitus [1]. Type 2 diabetes (T2D) is caused by a lack of insulin sensitivity in hepatic and peripheral tissues [2], combined with loss of insulin secretion due to decreased beta cell function and/or mass. Compensatory increases in insulin secretion precede T2D, and loss of this mechanism due to beta cell dysfunction or loss of beta cell mass, is a key factor in triggering frank manifestation of this disease [2,3].

### 1.1. Glucose-Stimulated Insulin Secretion from Pancreatic Beta Cells

Secretion of insulin from pancreatic beta cells is central in maintenance of whole-body glucose homeostasis (Figure 1). Glucose-stimulated insulin secretion (GSIS) occurs through a sequence of strictly controlled events, following a rise in blood glucose concentrations. Glucose is transported from the plasma across the cell membrane, via glucose transporter 1 (GLUT1) [4], where it becomes available for phosphorylation by free glucokinase (GCK) [5]. Glucokinase is activated by release from insulin granules following monomerisation of neuronal nitric oxide synthase (nNOS) [6,7,8]. The rise in glucose-6-phosphate increases glycolysis and aerobic respiration, and the resultant increase in ATP/ADP ratio causes closure of ATP-dependent K^+^ channels (K_ATP_) and membrane depolarisation, facilitating opening of voltage-gated calcium channels (Ca_V_). Influx of Ca^2+^ triggers exocytosis of insulin granules by interaction of insulin secretory granules with soluble *N*-ethylmaleimide-sensitive factor attachment protein receptor (SNARE) complexes located at the cell membrane, comprised of vesicle-associated membrane protein 2 (VAMP2), synaptosomal-associated protein 25 (SNAP25) and syntaxin-1A (STX1A) [9,10,11]. Following restoration of glucose homeostasis, the pancreatic beta cell membrane potential is corrected by voltage-gated potassium channels (K_V_2.1 and K_V_3.2 in human beta cells) [12].

### 1.2. Loss of Insulin Secretion and Beta Cell Mass in Diabetes

Insulin production by pancreatic beta cells exhibits a remarkable degree of plasticity (reviewed in [13]), responding acutely to differing environmental conditions including starvation and over-nutrition. However, the chronic, persistently high, demand for insulin, which occurs during insulin resistance, can lead to progressive dysfunction, and eventual loss, of beta cells [14,15]. The evaluation of GSIS ex vivo in pancreatic islets from T2D donors shows uncoupling between glucose concentration and insulin secretion: insulin secretion does not change at basal glucose concentrations, but exhibits reduced capacity to respond appropriately to rising glucose levels in patients with T2D [16,17,18,19,20]. Xenotransplantation of human T2D islets into immunodeficient diabetic mice was unable to restore normoglycaemia, unlike the implantation of non-diabetic islets [17], demonstrating that T2D limits beta cell function.

Multiple mechanisms contribute to dysfunctional insulin secretion in T2D beta cells. The expression of glucose transporters and glucokinase (GCK) is lower in human T2D islets than in islets of healthy controls, indicating impaired glucose sensing and metabolism [21]. Type 2 diabetes alters the expression of genes encoding proteins with a wide range of functions affecting insulin secretion, such as Ca^2+^ trafficking (*TMEM37*, *SUR1*), mitochondrial metabolism (*ALDOB*, *GPD2*, *FXYD2* and *PCK1*), cell cycling (*P21/CIP*, *TTC39C*) and fatty acid (*FFAR4*, *TMEM97*), insulin (*IR*), IGF-1 (*IGF1R*) and TNF (*TNFRSF11A*) receptor signalling [22,23,24,25,26,27]. There is also evidence of compromised mitochondrial function in T2D diabetic beta cells as a result of reduced enzymatic activity [16,28,29]. This can limit insulin secretion as it leads to a lower ATP/ADP ratio [30], and compromised Ca^2+^ influx. Protein and/or gene levels of SNARE complex, SNARE-modulating proteins syntaxin-1A, SNAP-25, VAMP-2, nSec1 (Munc), Munc 13-1, synaptotagmin V and synaptophysin, and components of the K_ATP_ channel (Kir6.2 and SUR1) were also lower in isolated pancreatic islets from diabetic patients, compared with controls, reflecting impaired insulin secretion in these individuals [31].

Loss of beta cell mass can occur during prediabetes [32]: at diagnosis, patients with T2D have often lost ~50% of their beta cell mass [33], from apoptosis and dedifferentiation [15,34,35]. Hyperinsulinaemia increases the production of hydrogen peroxide, activation of caspases and induces expression of inducible nitric oxide synthase (iNOS) and Tribbles homolog 3 (Trib3) in beta cells [36,37]. Prolonged exposure of rat islets or INS1-E cells to high concentrations of insulin leads to reduced phosphorylation of Akt^S473^, reductions in phosphorylation of P70S6 kinase and ERK-1/2 kinase and increased apoptosis [38]. Hyperglycaemia increases glyceraldehyde-derived advanced glycation end products (AGE) [39], while signalling via the receptor for AGE (RAGE) results in increased cytochrome release and caspase activation [40]. Exposure to elevated levels of glucose in combination with high levels of free fatty acids (glucolipotoxicity) is also thought to be a significant contributor to increased apoptosis and loss of GSIS in beta cells [41,42] (below).

De-differentiation is another factor contributing to loss of beta cell mass in T2D [43,44,45]. Transition of beta cells to a progenitor-like state or α cell [46,47] is associated with downregulation of beta cell identity genes, upregulation of beta cell ‘forbidden’ genes and upregulation of stem-cell genes [15,46]. John et al. (2018) observed downregulation of beta cell identity genes, *FoxO1*, *MafA* and *Nkx6.1*, in the *db/db* murine model of T2D [48], while genetic deletion of *FoxO1* in murine beta cells renders these cells more sensitive to metabolic stress, and is associated with upregulation of pluripotency genes such as *Ngn3*, *Oct4*, *Nanog* and *L-Myc* [49]. Similar findings were observed in islets from T2D patients, compared with healthy controls [43]. A larger proportion of T2D islets, compared with controls, showed a subpopulation of glucagon-positive cells that expressed cytoplasmic (inactive) FOXO1 and α-cell Aristaless-related homeobox transcription factor (ARX) [50], and a subpopulation of somatostatin-positive cells expressed cytoplasmic (inactive) homeobox protein NKX6.1 [43], suggesting dedifferentiation of beta cells and transition towards α- or δ-like cell physiology. Expression of the progenitor cell marker aldehyde dehydrogenase 1 family, member A3 (ALDH1A3) [51], is also observed in islets from T2D patients [43].

## 2. Lipid Accumulation and Beta Cell Dysfunction

It is established that over-accumulation of lipids and associated over-activation of lipid signalling pathways (lipotoxicity) contribute to loss of insulin secretion, beta cell toxicity and dysfunction, providing a link between obesity and T2D (reviewed in [41]). Glucolipotoxicity (GLT) describes the synergistic damaging effects of increased concentrations of free fatty acids (FFA) in the presence of high glucose concentrations (reviewed in [42]). Multiple outcomes are triggered in beta cells by GLT, including mitochondrial dysfunction and oxidative stress, endoplasmic reticulum (ER) stress and the protein unfolding response, inflammation and impaired autophagy, and loss of GSIS [42]. These changes reflect altered cell signalling pathways, increased expression of inflammatory cytokines, lipogenic and pro-apoptotic genes and proteins, and the accumulation of lipids, including diacylglycerols and triacylglycerols, ceramides, cholesterol and cholesteryl esters [42,52].

### 2.1. Fatty Acids, Diacylglycerols, Triacylglycerols and Beta Cell Dysfunction

The biosynthesis of triacylglycerol droplets, via intermediate diacylglycerols, is an important feature in many cell types, storing excess caloric intake against future need, and preventing the build-up of potentially toxic fatty acid derivatives [53]. Triacyglycerol synthesis occurs at the endoplasmic reticulum (ER), primarily from glycerophosphate and fatty acyl CoA. Diacylglycerol acyltransferase-1 (DGAT-1) plays an important role in esterifying (and thereby detoxifying) excess lipids entering the cell, while DGAT-2 esterifies fatty acids arising via de novo lipogenesis from glucose [53].

Fatty acid signalling plays an established (nutritional) role in stimulating insulin secretion by beta cells [54]: Jezek et al. (2018) recently reviewed the physiological roles of fatty acids in amplifying GSIS, inducing insulin granule exocytosis, and interacting with free fatty acid (FA) receptors [55]. The majority of in vitro studies examining the pathological impact of saturated FFA, such as palmitate and stearate, do so in the context of high glucose (GLT), in order to replicate diabetic conditions, although it is a difficult task to define the concentrations to which islets are exposed in vivo, as these depend on circulating levels, and factors influencing both delivery, uptake and release of FFA by islet cells [42]. Certainly, saturated fatty acids, in the presence of glucose, reduce insulin transcription by decreasing the expression of the transcription factor MafA, and translocation of pancreatic and duodenal homeobox 1 (PDX1), but these findings cannot be dissociated from the impact of GLT conditions on generation of ceramide (below) [42]. The molecular mechanisms by which palmitic acid induces apoptosis in beta cells are not completely understood, but may involve activation of kinases, including c-Jun N-terminal kinase (JNK), protein kinase C (PCK), p38 mitogen-activated protein kinase (p38MAPK), extracellular signal-regulated kinase (ERK) and Akt kinase pathways [56].

In contrast to saturated palmitic (C16:0) or stearic acids (C18:0), monounsaturated oleic acid (C18:1) is thought to improve beta cell survival and prevent loss of insulin signalling [57]; Cho et al. (2012) also demonstrated that arachidonic acid can protect against the damaging effects of palmitic acid in HIT-T15 pancreatic cells (loss of GSIS, DNA fragmentation and decreased cell viability) [58]. Notably, this protective mechanism was lost in the presence of a DGAT inhibitor, suggesting that the presence of the polyunsaturated fatty promoted sequestration of toxic palmitic acid into triacylglycerol [58]. Knockdown of fatty acid synthase (FAS), which decreases phospholipid and neutral lipid pools in INS-1 832.13 insulinoma cells, inhibits GSIS, suggesting that efficient storage of newly synthesised lipids is also important in sustaining insulin secretion [59].

Diacylglycerol, as a lipid signal messenger, has a physiological role in beta cells: its primary function is to activate protein kinase C (PKC0, but also triggers other pathways, such as the Munc-13-dependent pathway: the cellular level of diacylglycerol (DAG), which is tightly regulated by DAG kinases (DGK), acts as a positive regulator of insulin secretion [60]. However, diacylglycerol has also been shown to inhibit insulin release via a PKC-independent mechanism in HIT T-15 islet cells, via modulation of Ca^2+^ flux [61,62]. Sawatini et al. (2019) demonstrated a biphasic response to type I DGK inhibitor, R59949, in MIN6 β cells: while low concentrations of the type I DGK inhibitor, R59949, increase PKC-dependent insulin secretion, higher concentrations (>10 μM), which trigger higher levels of diacylglycerol, suppress this process, possibly via loss of voltage-dependent Ca^2+^ channel activity [63,64]. Esterification of both fatty acids and diacylglycerol into the relatively inert triacylglycerol pool protect against the accumulation of bioactive (and potentially toxic lipids) (above) [53,54,55]. Exposure of rat islets to elevated levels of glucose stimulates the formation of glycerol and fatty acids, and diversion of glucose carbons into triacylglcyerols and cholesteryl esters [65]. By contrast, elevated plasma concentrations of triacylglycerol-rich lipoproteins reflect increased fatty acid flux from adipose tissue, and are linked with diminution of insulin secretion and induction of insulin resistance in patient cohorts [66].

### 2.2. Ceramides and Sphingolipid Signalling in Beta Cell Dysfunction

A series of complex interactions, requiring both active synthesis and degradation, determine the cellular sphingolipid content (reviewed in [67]). Biosynthesis is initiated at the cytosolic face of the endoplasmic reticulum (ER), starting with the condensation of L-serine and palmitoyl CoA; reduction, acetylation and desaturation reactions result in the generation of ceramide, which acts as the central substrate for the production of other sphingolipid intermediates (reviewed in [68]). Hydrolysis of ceramide at the ER (neutral ceramidase), plasma membrane (alkaline ceramidase) and in the lysosome (acid ceramidase) generate sphingosine, which can be phosphorylated to sphingosine-1-phosphate by sphingosine kinase. Ceramide is transported from the ER to the Golgi, where is can be used to synthesise sphingomyelin and glucosylceramides; at the plasma membrane, ceramide kinase generates ceramide-1-phosphate (C1P), which can be hydrolysed back to ceramide by C1P phosphatase [68].

Dysregulated ceramide and sphingolipid metabolism has been linked with dysregulation of insulin secretion, and apoptosis of beta cells, in response to glucolipotoxicity and/or inflammatory cytokines. Veluthakal et al. (2009) demonstrated that the impact of palmitic acid under glucolipotoxic conditions can be mimicked by a cell-permeable ceramide analog which reduces the expression of nucleotide diphosphate kinase in INS832/13 cells [69], a feature which may contribute to abnormal G protein activation and impaired insulin secretion. Indeed, recent evidence implicates cross-talk between Ras-related C3 botulinum toxin substrate 1 (Rac1) and the ceramide signalling pathway in the onset of beta cell dysfunction [70]. Exposure to palmitic acid, in the presence of glucose, also impairs transcription of the insulin gene in MIN-6 cells, via activation of Per-Arnt-Sim kinase (PASK) and extracellular regulated kinases-1/2 (ERK1/2) [71].

Incubation with 0.4 mM palmitic acid under normoglycaemic conditions increases de novo synthesis of dihydrosphingosine and dihydroceramides in beta cells without inducing apoptosis; however, increasing the glucose concentration to 30 mM induced apoptosis, and amplified formation of C18:0, C22:0 and C24:1 (dihydro)ceramide species via upregulation of ceramide synthase 4 levels [72]. Activation of the extrinsic apoptotic pathway under glucolipotoxic conditions, mostly via initiator caspase 8, promotes apoptosis by cleavage and activation of downstream effector caspases like caspase 3 (reviewed in [73]). The lack of caspase 8 can protect against ceramide-induced beta cell death, and knockout of caspase-3 can protect mice against the development of autoimmune diabetes [74,75]. Other sphingolipid metabolites, including glycosphingolipids, sphingosine-1-phosphate and gangliosides, can affect beta cell signalling pathways, including apoptosis, cytokine release, ER to Golgi vesicular trafficking and insulin gene expression; the activity of neutral sphingomyelinases, which regulate the composition of the plasma membrane, can also affect beta cell excitability and insulin [76]).

### 2.3. Cholesterol Accumulation and Beta Cell Dysfunction

Effective cholesterol homeostasis in beta cells is an important factor in maintaining insulin secretion (reviewed in [77]). The uptake, synthesis and removal of cholesterol is tightly controlled by the functional opposition between the activities of sterol regulatory element-binding proteins (SREBPs) and liver X receptor (LXR α/β) transcription factors, while storage is facilitated by esterification to cytosolic droplets of cholesterol esters by acyl CoA: cholesterol acyl transferase (ACAT-1). The primary route for cholesterol uptake is via members of the low-density lipoprotein receptor (LDL-R) [78] and scavenger receptor families [79]. As the intracellular cholesterol content rises, SREBP-2 is sequestered (and inactive) at the endoplasmic reticulum, leading to loss of expression of genes encoding the LDL-R and the enzymes responsible for endogenous synthesis of cholesterol [80]. Instead, (oxy)sterol-mediated activation of nuclear LXR transcription factors, which form obligate heterodimers with retinoid X receptors (RXR), leads to induction of expression of genes encoding proteins involved in the ‘reverse’ cholesterol transport process, including ATP-binding cassette (ABC) transporters A1 (ABCA1) and ABCG1/G4 [81,82]). These transporters work in concert to remove excess cholesterol from cells, via efflux to (apo)lipoprotein acceptors such as apoA-I and high-density lipoprotein, respectively [82].

Naturally, the presence of excess cholesterol regulates the physical properties (fluidity, curvature and lipid raft content) of membranes that influence function and locale of membrane proteins such as receptors, ion channels and transporters, and vesicle formation and fusion, affecting several steps of the insulin secretory pathway [77] (Figure 2). Notably, glucose-stimulated insulin release is reduced by decreased glucose transporter activity [83], and stabilisation of the neuronal nitric oxide synthase (nNOS) dimer, which prevents the movement of glucokinase from insulin granules to the cytosol [77]. An increase in cellular cholesterol level can also increase plasma membrane-associated phosphatidylinositol 4,5 bisphosphate (PIP_2_) [84]: PIP_2_ dissociated from the plasma membrane is hydrolysed by phospholipase C leading to Ca^2+^ release from intracellular stores and may sensitise K_ATP_ channels leading to an influx of Ca^2+^ by Ca_V_ channels [85,86]. Alterations in the density of voltage-gated Ca^2+^ channels lead to reduced flux of Ca^2+^ into the beta cell, and decreased insulin secretion [87]. In addition, increased production of PIP_2_ activates dynamin, which acts to reduce full fusion events of granules at the plasma membrane [88,89], while accumulation of excess cholesterol in insulin granules causes dysfunctional retrieval of exocytosis proteins, such as clathrin, syntaxin 6 and vesicle-associated membrane protein 4 (VAMP4) [90]. These factors contribute to incomplete granule–membrane fusion, evidenced by longer duration and reduced lateral spreading of insulin granules [91].

Accumulation of cholesterol at the endoplasmic reticulum not only depletes calcium stores needed for insulin release [92,93] but can trigger protein unfolding by activation of the protein kinase RNA-like endoplasmic reticulum kinase (PERK)–phosphorylated eukaryotic initiation factor 2 alpha (eIF2α) [94] pathway, which results in global inhibition of protein synthesis (including preproinsulin) and translation of activating transcription factor 4 (ATF4) [95]. Build-up of sterol within the trans-Golgi network inhibits granule formation [90] while disruption of lipid rafts alters the sorting of granins, a key constituent of secretory granules, and of endoproteases, needed for the processing and maturation of the insulin hormone [96].

Conversely, reductions in cholesterol biosynthesis caused by ‘statin’ drugs, or the depletion of the plasma membrane cholesterol pool using methyl β-cyclodextrin (MCD), also inhibits GSIS and lowers insulin content in β cells and islets [97,98]. Depletion of cholesterol also affects the formation of insulin granules, while disruption of cholesterol-rich lipid rafts impairs insulin secretion by redistribution of SNARE (syntaxin and SNAP25) and K^+^ATP and voltage-gated Ca^2+^ channels [97,98]. High levels of glucose inhibit cholesterol biosynthesis, resulting in disruption of lipid rafts, redistribution of plasma membrane syntaxin 1A, loss of this protein from granule-docking sites, fewer docked granules and reduced insulin secretion [99]. Moreover, recent studies have suggested that use of statin drugs in dyslipidaemia can actually provoke new-onset diabetes in ‘prediabetic’ patients [77], and genetic variants in *HMGCR* have also been linked with predisposition to diabetes, again positing cholesterol biosynthesis as important in sustaining beta cell function [100].

## 3. Mechanisms Contributing to Changes in Gene Expression and Beta Cell (dys)function: microRNA

Over the last decade, mechanisms resulting in changes in gene expression, including chromatin modifications, DNA methylation, post-translational modifications of histones, and altered expression of non-coding RNA sequences, such as long non-coding RNA (lncRNA) and microRNA (miRNA/miR) have been implicated in regulation, and loss, of beta cell function and diabetes: a number of excellent reviews have recently covered these areas in depth [101,102,103,104,105,106].

MicroRNA sequences are small (~22 nucleotide) non-coding RNA sequences which regulate the expression of networks of genes in beta cells, in response to environmental factors such as caloric excess, obesity and diabetes [101,102,103,104,105,106]. These sequences can be isolated or clustered within the human genome, either between genes or within the intron–exon regions of genes encoding proteins [107,108]. Transcription of microRNA (miR) sequences is dependent on the expression and activity of RNA polymerase II/III [109,110], can be dependent or independent of mRNA expression [108,111,112] and occur via both canonical and non-canonical pathways [109,113]. In the canonical pathway, a hairpin-containing primary miRNA (pri-miRNA) transcript with a 5’-methylated cap and a 3’-polyadenylated tail is generated, which is then processed via a complex containing double-stranded RNA-binding protein DiGeorge syndrome critical region gene 8 (DCGR8) which recognises methyl motifs present in the pri-miRNA [114,115,116]. This interaction anchors Drosha, a ribonuclease III which generates precursor miRNA (pre-miRNA) by cleaving the hairpin structure from the pri-miRNA transcript [117,118,119]. The pre-miRNA (~70 nucleotides) are exported from the nucleus: exportin-5 interacts with the 3’ overhanging sequence of pre-miRNA, while RanGTP remains bound to the hairpin structure until hydrolysis of GTP to GDP in the cytosol results in release of pre-miRNA [120]. Cytosolic pre-miRNA is processed by Dicer (RNase III), which removes the stem–loop structure to generate a mature miRNA duplex (19–25 nucleotides in length) [121,122]. The guide strand is loaded onto the active RNA-induced silencing complex (RISC), made up of Dicer, TAR RNA-binding protein (TRBP) and argonaute (1–4) proteins; miRNA base pairs with their complementary mRNA molecules are guided by their miRNA recognition element [123,124].

A perfect (exact) or near-perfect complementary match between miRNA and the conserved 3’-UTR region of the target mRNA results in degradation of mRNA; if the complementarity is imperfect (partial), then moderate reductions in mRNA and translational repression occur [125,126,127,128]. The end result is decreased protein output from the target gene, albeit often quite modest in its magnitude [125,126,127,128], reflecting the role of microRNA in ‘fine-tuning’ gene and protein expression. Additional factors can reduce translational efficiency or induce mRNA destabilisation, including AU-rich regions near the ‘seed’-binding sites, auxiliary binding of miRNA to the target transcript, or mRNA deadenylation [127,128]. Each miRNA sequence can have target sites in hundreds of different genes, exhibiting differing degrees of complementarity: computational prediction suggests that >60% of all protein-coding genes are miRNA targets [125,129,130]. Tissue-specific and concentration-dependent effects are also noted, particularly in healthy tissues compared with pathological conditions [131,132,133,134]. Finally, some miRNA sequences exist in the extracellular environment, in microvesicles, like exosomes and ectosomes, complexed with proteins, or transported in lipoproteins such as HDL, and have been widely employed as biomarkers of health and disease [135,136,137,138].

### 3.1. MicroRNA Sequences Linked with Lipid Accumulation in Beta Cells

Over the last decade, it has become clear that the network of genes encoding proteins involved in lipid metabolism and cholesterol homeostasis also lies under the control of microRNA sequences, such as miR-33 [139]. Table 1 lists some of the microRNA sequences, derived from interrogation of the NCBI/PubMed database which are altered by changes in metabolism induced in beta cells and islets. It is clear that multiple miRNA sequences are regulated in beta cells by exposure to metabolic challenges, targeting an array of genes and processes involved in beta cell function. In particular, induction of miR-34a is strongly linked with beta cell lipotoxicity associated with exposure to saturated fatty acids in vitro and in vivo, via multiple mechanisms [140,141,142,143,144] (Table 1), which may also reflect increased flux of fatty acids through the diacylglycerol/triacylglycerol, ceramide/sphingolipid and cholesterol esterification pathways. These include targeting sirtuin 1 (SIRT1), an NAD^+^-dependent deacetylase, which activates expression of tumour-suppressor protein p53, DNA repair factor Ku70, nuclear factor κB (NF-κB), STAT3 and the FOXO family of forkhead transcription factors [145]. Sirtuin 1 aids suppression of cellular senescence, delays age-related telomere attrition, promotes DNA damage repair and cell survival, and reduces apoptosis [145]; loss of this protein, due to elevation of miR-34a after exposure to saturated fatty acids, is therefore entirely consistent with enhanced lipotoxicity in β cells. MiR-34a also directly targets lactate hydrogenase, thereby repressing the increased glycolysis observed in proliferating cancer cells (reviewed in [146]), and targets peroxisome proliferator activator receptor α (PPARα) in liver cells [reviewed in 147], both of which may impact utilisation of fatty acids; whether these factors contribute to toxicity in beta cells remains unknown. Certainly miR-34a, itself a target of p53, is an established tumour suppressor, and repression or dysregulation of this sequence is noted in a number of human cancers, leading to the development of MRX34, a liposomal miR-34a mimic, as a putative therapeutic (discussed further below) [146,147,148].

Other microRNA sequences altered in beta cells by exposure to saturated fatty acids, and linked with lipotoxicity, include miR-146a [140,141], miR-182-5p [149], miR-297b-5p [150,151], miR-374c-5p [151], miR-375 [152] and miR-3074-5p [153] (Table 1). MicroRNA-146 exists in two forms (miR-146a/b), often not distinguished despite their distinct chromosomal locations [154], but which share a seed region and target some of same genes involved in the host immune response and inflammation, such as Toll-like receptors. The role of ‘mirR-146’ in promoting apoptosis appears context dependent: mir-146a-5p promotes the apoptosis of chrondrocytes via activation of the NF-κB pathway [155], while miR-146b enhances apoptosis of gastric cancer cells by targeting protein tyrosine phosphatase 1B (PTP1B) [156]; by contrast, ‘miR-146’ protects against cardiomyocyte apoptosis by inhibiting NF-κB [157], and blocks the pro-apoptotic and inflammatory effects of lipopolysaccharide (LPS) in lung cancer cell lines [158]. Fred et al. (2010) also demonstrated that in human islets, the level of ‘miR-146’ increases after exposure to pro-inflammatory cytokines, decreases after culture in media containing high glucose, but was not changed by exposure to palmitate [159].

Notably, both miR-146b and miR-182-5p have been linked with protection against high-fat diet-induced non-alcoholic steatohepatitis in mice: exposure to miR-146b reduces the expression of IL-1 receptor-associated kinase (IRAK1) and tumour necrosis factor (TNF) receptor-associated factor 6 (TRAF6) after exposure to oleic acid, reducing inflammation and lipid accumulation in vitro and in vivo [160]. In the same models, miR-182-5p reduced oleic acid-induced hepatic expression of TNFα, IL-6 and TLR4 [161]; this sequence also prevents apoptosis, and reduces the levels of cluster of differentiation (CD) 36, total cholesterol and triglyceride in macrophages after exposure to oxidised LDL, again by targeting TLR4 [162]. In β cells, miR-182-5p directly targets thrombospondin-1 (TSP-1) [149], a CD36 ligand, which, in human hepatic cells, regulates lipid metabolism by inhibiting the proteolytic cleavage of SREBP-1, reducing lipogenesis and triglyceride accumulation [163]. However, genetic deletion of TSP-1 in mice is associated with reduced plasma lipid levels and hepatic inflammation, and activation of PPARα [164], and decreased obesity-induced microvascular complications in apoE^−/−^ mice [165], findings which resonate with the impact of miR-182-5p in beta cells (Table 1).

One other study remarks the impact of miR-374c-5p on apoptosis, controlling the proliferation, migration, epithelial-mesenchymal transition and apoptosis of human breast cancer cells, via repression of TATA box-binding protein associated 7 (TAF7) and expression of DEP domain containing 1 (DEPDC1), a transcriptional co-repressor involved in the promotion of carcinogenesis [166]. The tumour suppressive sequence miR-3074-5p has also been linked with increased apoptosis in both trophoblasts and breast cancer [167,168], but no direct reports link this sequence with altered lipid metabolism. By contrast, a number of recent studies cite both positive and negatives roles for miR-375 in apoptosis, of chrondocytes, breast, colon, gastric and hepatic cancer cells and cardiomyocytes [169,170,171,172,173,174,175,176]. In mice (C57BL/6), miR-375 blocks high-fat diet-induced insulin resistance and obesity, by inhibiting over-activation of the aryl hydrocarbon receptor and promoting hepatic expression of genes involved in responses to insulin [175], providing protection against the high-fat diet. Notably, this sequence is thought to play an established role in beta cell function and in diabetes: (reviewed in [176]). During the development of the pancreas, the increased expression of miR-375 parallels increased expression of the insulin gene, and proliferation of β cells, while loss-of-function (LOF) knockdown of this sequence in zebrafish and mice suggests a key role in determining the balance between β cells (↓) and α cells (↑). MicroRNA-375 is thought to inhibit GSIS via a number of mechanisms, including targeting myotrophin (Table 1) [152], pyruvate dehydrogenase kinase-1 (PDK1), PI3-kinase and interactions with cAMP-directed pathways [176].

Finally, exposure to saturated fatty acids lowers the expression of miR-297-5p in vitro and ex vivo [150,151], which targets large-tumour-suppressor kinase 2 (LATS-2) [150]. This kinase is a central regulator of cell fate, influencing the function of a host of oncogenic or tumour suppression factors, and is a core component of the canonical Hippo pathway [177], so increased expression of LATS-2 is consistent with the lipid-induced apoptosis seen in beta cells (Table 1). Notably, LATS-2 inhibits the activation of SREBP-2, and suppresses cholesterol accumulation in hepatic cells [178]: if replicated in beta cells, enhanced LATS-2 expression could therefore additionally promote cholesterol deposition. MiR-7222-3p, which is also elevated by exposure to palmitate, targets ACAT-1: loss of this protein would enhance the level of potentially toxic-free cholesterol in beta cells, by abrogating storage as neutral cytosolic droplets of cholesteryl ester [179].

Of the sequences directly moderated by cholesterol exposure in beta cells and islets, elevations in two (miR-27a and miR-33a) [180,181] are linked with repression of ABCA1; it is well established that this cholesterol efflux transporter, its apolipoprotein acceptors (e.g., apoA-I, apoE) and its product, HDL itself, can provide protection to beta cells and pancreatic islets [182,183,184,185,186,187,188] and in experimental models [187,188]. Some, but not all, clinical studies also provide support for this concept [189,190,191]. These protective functions have been linked with the removal of excess cholesterol from beta cells, while others cite sterol- and/or transporter independent effects of apoA-I and HDL [192,193,194,195,196,197]. Certainly, miR-33a is one of the most intensively studied miR sequences in lipid metabolism and is integral to these responses [182,198,199]. Mir-33a is encoded in an intronic region of *SREBF2* and thus forms a regulatory link between the active expression of this gene, and repression of ABCA1 and ABCG1: knockout of miR-33a in mice promotes cholesterol efflux via ABCA1 and ABCG1, increases circulating levels of HDL and hepatic excretion of cholesterol in bile (reviewed in [197]), and this sequence is currently under exploration as a possible clinical target [198,199]. Notably, ABCA1 is also regulated by exposure to elevated levels of glucose: miR-145 increases the total level of islet cholesterol [200], while miR-383 targets the anti-inflammatory Toll-like receptor 4 [201], which has also been linked with altered expression of ABCA1 [182].

More complex relationships between miRs and HDL emerge from Table 1: Tarlton et al. (2021) showed that miR-21-5p could mimic the effects of HDL on targets STAT3 and SMAD7, but could not provide equivalent protection against glucolipotoxicity in human PANC hybrid 1.1B4 cells [202], while HDL increases the export of miR-375-3p, a feature which correlates inversely with insulin secretion in murine MIN6 cells [203]. Cholesterol exposure also enhances the expression of miR-24a, which impacts on the transcription factor Sp1 to alter the expression of secretagogin and reduce insulin secretion [204]. This sequence has been closely linked with changes in lipid metabolism in other cells and tissues: obesity induces overexpression of miR-24 [205], a sequence which associates with HDL, and that regulates cholesterol uptake by targeting scavenger receptor (SR) B-1. The same sequence enhances atherosclerosis by reducing lipid uptake from HDL via repression of SR-B1 [206] and, intriguingly, can control triacylglycerol biosynthesis by targeting fatty acid synthase [207]; inhibition of miR-24 can also help to limit hepatic lipid accumulation and hyperlipidaemia [208], suggesting this sequence may integrate neutral lipid metabolism in beta cells.

### 3.2. MicroRNA Biomarkers: Associations with Changes in Lipid Metabolism in Beta Cells

Some of the sequences altered by metabolic challenges in beta cells have also emerged as biomarkers of diabetes in the circulation (Table 2), although the epigenomic landscape in the bloodstream, as in cells and tissues, is obviously much more complex. Approximately 10% of miRNAs are thought to be secreted encapsulated in exosomes, with the remainder stably complexed with proteins such as argonaute 2 and nucleophosmin 1, and with HDL, under vesicle-free conditions; all of these forms protect miRNA against RNase degradation, allowing their delivery to recipient cells and tissues, and promoting intercellular communication [209]. MicroRNA sequences are thought to be selectively secreted into extracellular vesicles, just as the proteomic profile of secreted exosomes differs from parental cells [210], although routine analysis of the RNA content of extracellular vesicles in liquid biopsies is a challenging proposition [211]. Intriguingly, inhibition of neutral sphingomyelinase-2, which is rate limiting for synthesis of ceramides (above) decreases the amount of miRNA in exosomes (but not parent cells) [212]; ceramide synthesis is also thought to be involved in the functionally distinct, and possibly opposing, pathway mediated by HDL [212].

The data in Table 2 describe the outcome of searches for circulating miRNA sequences in studies relating to diabetic patients [213,214,215,216,217,218,219,220,221,222,223,224,225,226,227,228,229,230,231,232,233,234,235,236,237,238,239,240,241,242,243,244,245,246,247,248,249,250,251,252,253,254,255,256,257,258,259,260,261,262,263,264,265,266,267,268,269,270,271]; of these, 21 studies identify at least one of the sequences described in Table 1. It is clear that the directions of change of such biomarkers are not always consistent between differing studies (Table 2) or indeed, when comparing biomarker studies with outcomes from cells and tissues (Table 1 vs. Table 2). The sequences identified as biomarkers in Table 2, which have also been linked with changes in lipid metabolism in beta cells and islets, include miR-21 [216,244,256,257,261,270], miR-24 [216,229,230,257,261], miR-27a [218], miR-34a [213,229,261,270], miR-145 [247], miR-146a [218,219,228,239,241,244,245,265,266], miR-182 [210,218,237] and miR-375 [217,218,228,243,253]. Hsa-miR-146a emerges as one of the most common regulated sequences linked with diabetes, with decreased levels seen in samples derived from the blood of patients with diabetes [218,219,241,245,265,266] and those with diabetic foot and nephropathy [228], although Mensa et al. (2019) observed increased levels seen in diabetic women, compared with diabetic men [219]. Elevated levels of this sequence were also found in gingival crevicular fluid [239] and corneal samples [244] of type 2 diabetic patients, while in murine beta cells, and *db/db* islets, levels of miR-146a are increased in response to exposure to palmitate [140], predicating apoptosis (Table 1). Levels of hsa-miR-34 decreased in the bloodstream in two studies of diabetic patients [213,229], but increased in the report from Seyhan et al. (2016) [270]; levels also increased in murine cells exposed to palmitate [141,142,143,144]. These differing outcomes may reflect the progression of the disease, or the selective retention of this sequence under pro-apoptotic conditions. Clearly, there are challenges remaining in relating complex outcomes in cells and tissues with the epigenetic profile found in fluid biopsies.

### 3.3. Predictive Analysis (DIANA/KEGG) of Pathways Implicated in Beta Cell Dysfunction in the Face of Metabolic Challenges

Bioinformatic analysis was carried out on microRNA sequences identified as associated with lipid metabolism in pancreatic beta cells using miRPath v3.0 [272]. Full details of the list of miRs can be found in the Mendeley dataset deposit (doi: 10.17632/jnz8h974gc.1). In brief, all miRNAs named in Table 1 were used, sequences described in Table 1 were verified in miRbase before inclusion: where 5p or 3p sequences were not specified, both were included in the search list (full search list: hsa-miR-21-5p, hsa-miR-24-1-5p, hsa-miR-24-2 5p, hsa-miR-27a-3p, hsa-miR-33a-3p, hsa-miR-33a-5p, hsa-miR-34a-3p, hsa-miR-34a-5p, hsa-miR-124-3p, hsa-miR-124-5p, hsa-miR-145-3p, hsa-miR-145-5p, hsa-miR-146a-3p, hsa-miR-146a-5p, hsa-miR-182-5p, hsa-miR-297-5p, hsa-miR-374c-5p, hsa-miR-375, hsa-miR-383-3p, hsa-miR-383-5p and hsa-miR-3074-5p). Figure 3A demonstrates the Kyoto Encyclopaedia of Genes and Genomes (KEGG) pathways with a *p* < 0.01 association with the miRNA sequence involved in lipid metabolism in pancreatic beta cells. This recognises several lipid pathways (fatty acid biosynthesis, fatty acid metabolism, fatty acid elongation, glycosphingolipid biosynthesis and biosynthesis of unsaturated fatty acids) which validates the principle underlying the search. The additional pathways are indicative of the pleiotropic nature of miRNAs which have multiple gene targets, and highlight the importance of lipid homeostasis to beta cell function and health, with pathways implicated that affect pancreatic islet architecture and morphology (ECM-receptor interactions, Hippo signalling pathway, adherens junctions, thyroid hormone signalling pathway), mitosis (cell cycle), cellular energy homeostasis (FoxO signalling pathway) and cell survival (p53 signalling pathway); miRNA-pathway interactions in Figure 3B were mapped in Cytoscape v3.8.0 [273].

### 3.4. Therapeutic Applications of microRNA (targets) in Beta Cells

MicroRNA pathways can be regulated pharmacologically, and treatments involving miRNA focus on influencing dysregulated levels of miRNA in disease, including suppression or enhancement of key sequences [133]. Gene silencing can be achieved using artificial, double-stranded RNA fragments (mimics) that bind to target mRNA, resulting in activation of the RISC complex, downregulation of specific mRNA, and gene suppression. Synthetic oligonucleotides can bind to mature miRNA targets, leading to reduced RISC activity and upregulation of specific mRNA and protein expression; target ‘mimicry’ can also employ miRNA sponges, masking or erasers [133].

The practical utility of these approaches is exemplified by Miraversen (miR-122) which effectively reduced the expression of hepatitis C virus in Phase II clinical trials (2017) without significant side-effects [274,275,276], and by MRX34, a lipsomal miR-34a mimic, which entered Phase I clinical trials for treatment of advanced liver cancer [277,278]. However, the trial of the latter was halted by the Food and Drug Administration (FDA) in 2016, due to severe immune-mediated toxicity and four patient deaths (reviewed in [141]). Contributing factors may include the packaging vehicle, which was not designed to specifically target the miRNA to cancer cells, or the dose or dose schedule: at present, the reasons for the immune-related adverse events are not understood, and were not predicted by preclinical studies in animals, including non-human primates [141].

The packaging vehicle is key to efficient gene regulation as they facilitate passage through many physiological barriers before reaching the target tissue; options include metal, polymer or lipid nanoparticles, liposomes and hydrogels [279]. Packaging vehicles can also be designed to improve delivery of miRNAs to target tissues: some examples include the modification of polyamidoamine (PAMAM) nanocarriers with folic acid to direct them to cancers that overexpress folate receptors, and an amino acid sequence on cationic liposomes which increased delivery of siRNA to osteogenic cells [280,281]. Ensuring a targeted approach is important to limit off target effects. For example, one plausible target to protect beta cell survival is miR-34a (Table 1); however, this sequence is also a tumour suppressor. Thus, any therapeutic based on targeting miR-34a would require a delivery system directed specifically to beta cells, to avoid global targeting that could lead to uncontrolled growth in healthy tissue leading to cancer [282,283,284]. Several therapeutics are currently being developed that target miRNAs associated with lipid metabolism in beta cells (Table 1) including miR-21, miR-145 and miR-146a; however, none of them are designed to deliver specifically to the pancreas, or indeed, as T2D treatments [285,286,287]. Delivery of miR-216a to the pancreas using nanoparticles has been achieved in vivo [288]; while the nanoparticles were not designed to target the pancreas and improve delivery, the study demonstrates that nanoparticles can enter the pancreas and accumulate therein.

Thus, a number of factors must be considered in developing miRNA-based therapeutics, not least the number of target genes and cell signalling networks affected by these sequences [289], but also effective (tissue-specific) vectors and delivery processes [279,280,281,282,283,284]. Consideration of miRNA networks may also be useful when examining how to make effective therapies from miRNAs [202,289]. Designing treatments that comprehensively alter miRNA networks may achieve improved outcomes while retaining specificity by targeting a specific network. Another issue is that beta cells can export miRNAs which can then modulate gene expression in recipient tissues, which may complicate attempts to limit effects to the pancreas [138,203]. Careful scrutiny of these factors may facilitate the development of new drugs that can provide new treatment options for T2D patients that are more specific and safer than currently available options.

## 4. Conclusions

The explosion of interest in factors regulating gene expression, and beta cell function, over the last decade has revealed networks of genes regulated by multiple microRNA sequences, and the discovery of new pathways contributing to type 2 diabetes. This review has focused on miRNA sequences which are altered by changes in lipid metabolism in beta cells and islets, and highlighted the pleiotropic roles of these sequences in protecting against apoptosis or exacerbating lipid accumulation in these cells and tissues. Ultimately, the development of research in this field may lead to RNA-based therapeutics capable of sustaining beta cell function and preventing progression to type 2 diabetes.

## Figures and Tables

**Figure 1 biology-10-00534-f001:**
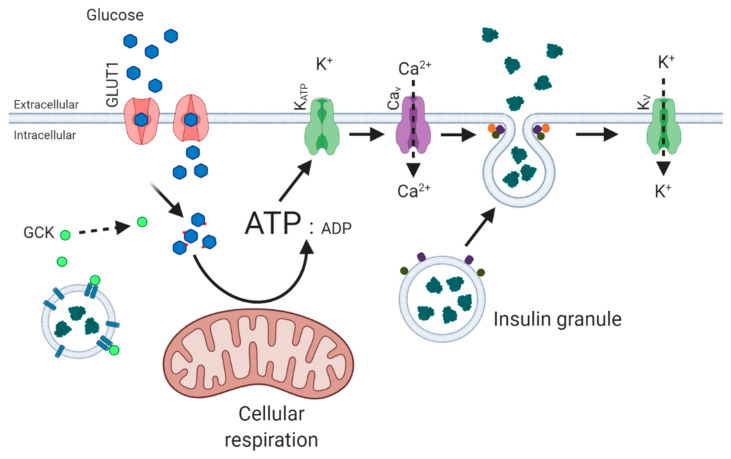
Glucose-stimulated insulin secretion from pancreatic beta cells. Glucose is imported via GLUT1 and phosphorylated by GCK. The increase in cellular respiration leads to an increase in the ATP/ADP ratio and the closure of K_ATP_ channels and opening of Ca_V_ channels. The entry of Ca^2+^ ions prompts the fusion of insulin granules with the cell membrane, and this process is facilitated by SNARE complexes. Feedback signals after the return to glucose homeostasis leads to restoration of the membrane potential through the influx of K^+^ ions through K_V_ channels.

**Figure 2 biology-10-00534-f002:**
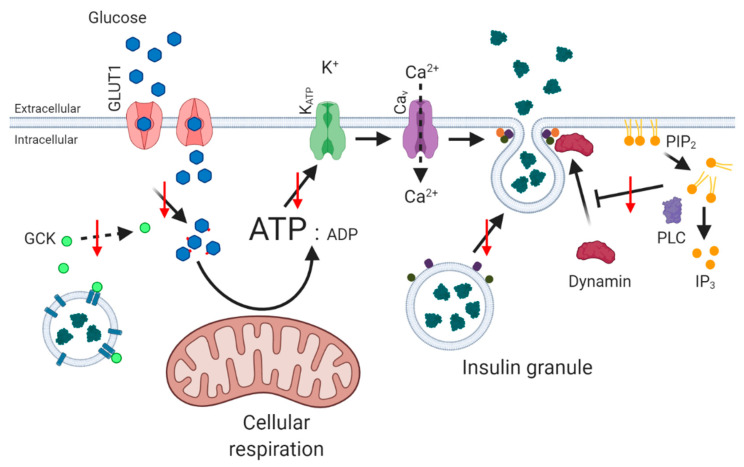
Detrimental effects of cholesterol accumulation on GSIS. High intracellular cholesterol has been shown to negatively affect GSIS through many processes including decreased glucose transporter activity, reduced active GCK leading to decreased change in ATP/ADP ratio, less insulin granule fusion partially due to an increased dynamin activity.

**Figure 3 biology-10-00534-f003:**
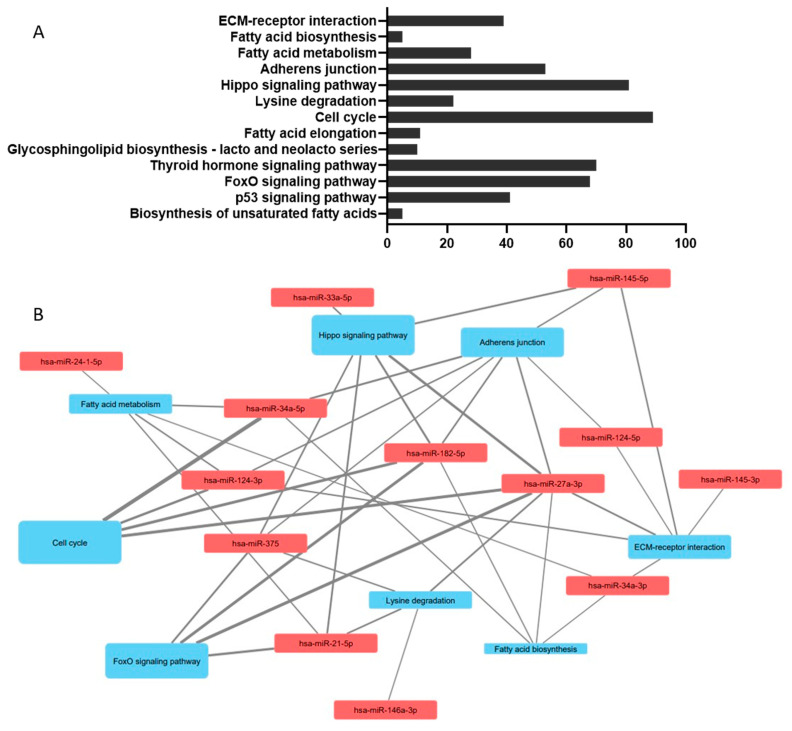
(**A**) The KEGG pathways identified through miRPath v3.0 strongly associated with miRNAs involved in lipid metabolism in pancreatic beta cells. All miRNAs included in the analysis were conserved between species with two caveats: mature rno-miR-182 shares 95% sequence similarity with hsa-miR-182-5p and mmu-miR-297b-5p shares 71% sequence similarity with mature hsa-miR-297b. Disease-specific pathways were excluded from the results. Pathways are ranked in order of the most closely linked pathways. (**B**) Interaction network between miRNAs involved in lipid metabolism in pancreatic beta cells mapped in Cytoscape v3.8.0. The network is annotated so that the size of the pathways labels is proportional to the number of genes in the network associated with the relevant miRNAs and the thickness of the edges shows the contribution of that miRNA as a proportion of the total miRNA–gene interactions in each pathway. Pathways associated with <4 miRNAs were excluded from the analysis to improve clarity of the map.

**Table 1 biology-10-00534-t001:** MicroRNA sequences altered in response to metabolic challenges in islets and beta cells. Upregulation (↑) or downregulation (↓) is noted next to the microRNAs.

MicroRNA (↑↓)	Cell/Tissue	Test Condition	Target	Outcomes	Reference
miR-21-5p (↑)	Human PANC hybrid 1.1B4 pancreatic beta cell line	High-density lipoprotein (HDL)	Signal transducer and activator of transcription 3 (STAT3) and decapentapegic protein 3 (SMAD7)	HDL protects against glucolipotoxicity (GLT); miR-21-5p mimic, replicates HDL repression of SMAD7 and STAT3, but does not protect against GLT.	Tarlton et al. (2021) [202]
miR-24 (↑)	Murine MIN6 pancreatic beta cell line	Cholesterol exposure	Transcription factor specificity factor 1 (Sp1) and Ca^2+^ sensor, secretagogin (Scgn)	Impaired Scgn-mediated phosphorylation of focal adhesion kinase and paxillin; reductions in focal adhesions in insulin granules and insulin secretion.	Yang et al. (2019) [204]
mir-27a (↑)	Rat INS-1 beta cell line	Cholesterol exposure	ATP-binding cassette transporter A1 (ABCA1)	GLP-1 ↓miR-27a, and increases expression of ABCA1: protects against cholesterol-induced lipotoxicity.	Yao et al. (2017) [180]
miR-33a (↑)	Islets isolated from mice with beta cell expression of human islet amyloid polypeptide (IAPP)	Cholesterol exposure	ATP-binding cassette transporter A1 (ABCA1)	↑miR-33 increases IAPP deposition; knockout of beta cell ABCA1 in hIAPP ^+/−^ mice impairs glucose tolerance, insulin secretion, induces hyperglycaemia.	Wijesekara et al. (2016) [181]
miR-34a (↑)	Murine MIN6B1 beta cell line Pancreatic islets (*db/db*) mice	Palmitate exposure	Tumour protein p53 Vesicle-associated membrane protein 2 (VAMP-2)	↑ miR-34a causes sensitisation to apoptosis, and reduces insulin secretion. Anti-miR-34a oligonucleotide partially protection against apoptosis.	Lovis et al. (2008) [140]
miR-34a (↑↓)	Rat INS-1 islet beta cell line	Glucagon-like peptide-1 (GLP-1) and palmitate exposure	Sirtuin 1	Palmitate exposure ↑ miR-34a; GLP-1 ↓ miR-34a; miR-34a mimics enhances palmitate lipotoxicity; inhibitors achieve the reverse.	Han et al. (2012) [141]
miR-34a (↑)	Murine MIN6B1 pancreatic beta cell line	Palmitate exposure	B-cell lymphoma cell-2 (Bcl-2)	MiR-34a interacts directly with Bcl-2; mir-34a mimic promotes lipoapoptosis; inhibitor achieves the reverse.	Lin et al. (2014) [142]
miR-34a-5p (↑)	Murine islets and rat INS-1 insulinoma cells	Stearic acid	B-cell lymphoma cell-2 (Bcl-2)	Lipotoxicity reduced by inhibitors of miR-34a-5p; protein kinase-like endoplasmic reticulum kinase (PERK) and p53 mediate stearic acid elevation of miR-34a-5p.	Lu et al. (2016) [143]
miR-34a (↑)	Rat INS-1 islet beta cell line	Exposure to ‘high’ glucose and palmitate (GLT)	Sirtuin 1 (SIRT1)	Long non-coding (lnc)RNA, NONRATT003679.2, ‘sponges’ miR-34a and reduces cell injury due to GLT.	Kong et al. (2019) [144]
miR-145 (↓)	Murine pancreatic islets	Elevated glucose	ATP-binding cassette transporter A1 (ABCA1)	↑miR-145 increases total islet cholesterol, and decreases GSIS; inhibitors achieve the reverse.	Kang et al. (2013) [200]
miR-146a (↑)	Murine MIN6B1 pancreatic beta cell line Pancreatic islets (*db/db*) mice	Palmitate exposure	-	↑ miR-146a induces sensitisation to apoptosis. Anti-mir-146a oligonucleotide provides partial protection against apoptosis.	Lovis et al. (2008) [140]
miR-182-5p (↑)	Rat INS-1 beta cell line	Palmitate exposure	Thrombospondin 1 (THBS-1)	Mimic of mir-182-5p decreased viability and increases lipotoxicity due to palmitate; inhibitor achieves the reverse.	Liu et al. (2018) [149]
miR-297b-5p (↓)	Murine TC6 beta cell line. Islets derived from C57BL/6 mice	Stearic acid or palmitic acid exposure	Large-tumour-suppressor kinase 2 (LATS2)	Upregulation of miR-297b-5p protects against stearic acid-induced apoptosis and loss of insulin secretion.	Guo et al. (2020) [150]
miR-297b-5p (↓)	Murine TC6 beta cell line	Stearic acid exposure	-	-	Yu et al. (2020) [151]
miR-375 (↑)	Murine TC6 beta cell line	Stearic acid exposure			Yu et al. (2020) [151]
miR-375 (↑)	Murine NIT-1 cells	Palmitate exposure	Myotrophin (V1) protein	↑ miR-375 increases susceptibility to palmitate-induced lipoapoptosis; knockdown of endogenous pri-miR-375 protects against lipoapoptosis	Li et al. (2010) [152]
miR-375-3p	Human islets, INS-1 and MIN6 bet cell lines	HDL	-	Export of miR-375-3p to HDL correlates inversely with insulin secretion.	Sedgeman et al. (2019) [203]
miR-383	Murine MIN6 beta cells	Exposure to ‘high’ glucose	Toll-like receptor 4 (TLR4) ApoC3	Overexpression of miR-383 inhibits glucose-dependent apoptosis and oxidative stress.	Cheng et al. (2020) [201]
miR-3074-5p (↓)	Rat INS-2 cells and murine islets	Palmitate exposure	cAMP-responsible element-binding protein (CREB)	Long non-coding RNA lncEif4g2 ↓miR-3074-5p, decreases CREB, enhances nuclear factor erythroid 2-related factor 2 (Nrf2) and protects against lipotoxicity.	Wang et al. (2020) [153]
miR-7222-3p (↑)	Murine MIN6 pancreatic beta cell lines	Palmitate exposure	Acyl CoA: Cholesterol Acyltransferase (SOAT1)	Circular RNA circ-Tulp4 promotes beta cell function by sponging miR-7222-3p and regulating SOAT1.	Wu et al. (2020) [179]

**Table 2 biology-10-00534-t002:** Circulating biomarkers of diabetes, cross-referenced to microRNA sequences modified by metabolic challenges in beta cells. Upregulation (↑) or downregulation (↓) is noted next to the microRNAs.

MicroRNA in T2D (↑↓)	Study Design	Subjects	Groups	Specific Association Tested	Reference
39 regulated miRs identified: hsa-miR-34a (↓) hsa-miR-182 (↑)	Measured levels of 170 miRNAs in plasma	Women with obesity and sex-matched controls	T2D (15), insulin resistance (IR) (19), insulin sensitivity (IS) (12) and controls (12)	T2D patients with obesity compared to controls	Jones et al. (2017) [213]
9 regulated miRs identified: no matches with Table 1	Whole peripheral blood: RNA sequencing prior to weighted gene coexpression network analysis (WGCNA)	40–60-year-old men	Newly diagnosed T2D (3) and control (3)	T2D patients compared to controls	Feng et al. (2019) [214]
6 regulated miRs identified: no matches with Table 1	Measured 179 microRNA sequences in exosomes isolated from serum	Men matched for age and BMI	Control (NGT) (4) and T2D (4)	T2D patients compared to controls	Katayama et al. (2018) [215]
30 regulated miRs identified, including hsa-miR-21 hsa-miR-24	Measured 754 small noncoding RNAs in pooled samples isolated from serum	Subjects selected from the Bruneck study	Diabetes patients (10) and age, sex and risk factor profile-matched controls (30)	T2D patients compared to controls	Zampetaki et al. (2010) [216]
4 regulated miRs identified, including hsa-miR-375 (↓)	Four miRNAs measured	Subjects initially non-T2D at start of CORDIOPREV (cardiovascular disease) study	Controls (78), prediabetic (223), incident prediabetic (30) and incident T2D (107)	T2D patients at risk of CVD compared to controls at risk of CVD	Jiménez-Lucena, Carmago et al. (2018) [217]
82 regulated miRs identified, including hsa-miR-27a (↑) hsa-miR-146a (↓) hsa-miR-182 (↓) hsa-miR-375 (↑)	MicroRNAs from miRbase 11.0 were measured from whole blood samples	Men with minimal differences in their clinical characteristics between the groups	Controls (7), impaired fasting glucose (IFG) (6) and T2D (8)	T2D patients compared to Controls	Karolina et al. (2011) [218]
hsa-miR-146a (↓) in males hsa-miR-146a (↑) in females	Circulating miR-146a levels in serum	Subjects recruited from the Italian National Research Center on Aging (INRCA)	Controls (188) and T2D patients (144)	T2D patients compared to controls	Mensà et al. 2019 [219]
No match with Table 1	Circulating miR-135a levels in plasma	Case–control study; T2D patients were treatment naive	Controls (40), prediabetic patients (40) and T2D patients (40)	T2D patients compared to controls	Monfared et al. (2020) [220]
No match with Table 1	Circulating miR-30c	Patients that had previously undergone coronary artery angiography	Control (32), coronary heart disease (CHD) (34), non-complicated T2D (47) and CHD + T2D (27)	T2D patients compared to controls	Luo et al. (2019) [221]
No match with Table 1	Circulating miR-103a and miR-103b	Patients that had previously been assessed at the Department of Endocrinology, Alliliated Hospital of Southwest Medical University, Luzhou, Sichuan, China	Controls (50), prediabetes (47) and T2D (48).	T2D patients compared to controls	Luo et al. (2020) [222]
No match with Table 1	Measured 179 miRNAs most highly abundant in human serum/ plasma in microvesicle isolations	Patients selected from primary health care on the Evolution of Patients with Prediabetes (PREDAPS) study. No patients had T2D at the start of the study	Control (8), fatty liver (8), prediabetes (8) and T2D (7)	T2D patients compared to controls	Parrizas et al. (2020) [223]
Four regulated miRs identified: no matches with Table 1	PBMC sample RNA sequencing for bioinformatic analysis	Recruited from Outpatient Clinics of Division of Endocrinology, Ribeirão Preto Medical School, University of São Paulo and of the São Paulo Federal University, Brazil. Controls were matched to patients	Controls (40), T1D (31) and T2D (32)	T2D patients compared to controls	Massaro et al. (2019) [224]
No match with Table 1	Abundance of miR-128 was measured in serum	Recruited from the Chennai Urban Rural Epidemiology Study (CURES)	Control (40), depression patients (40), T2D (40) and T2D with depression (40)	T2D patients compared to controls	Prabu et al. (2020) [225]
No match with Table 1	Measured 325 miRNAs expression in plasma samples	Patients admitted in Lanzhou University Second Hospital. Controls were matched to patients	Control (40), T2D patients with norm-albuminuria (40) and T2D patients with albuminuria (66)	T2D patients compared to controls	Wang et al. (2019) [226]
Four regulated miRs identified: No matches with Table 1	Profiled 752 miRs isolated from PBMCs	Patients recruited with abdominal aortic aneurysm (AAA)	Control (3) and T2D (3)	T2D patients with AAA compared to controls with AAA	Lareyre et al. (2019) [227]
Two regulated miRs identified: hsa-miR-146a (↓) hsa-miR-375 (↓)	Circulating miR-146a, miR-34a and miR-375 in serum	Cross-sectional study conducted in Mexico	Control (35), prediabetes (16), T2D (54), T2D with nephropathy (18), T2D with diabetic foot (3) and T2D with CVD	T2D patients with diabetic foot or nephropathy compared to controls	Garciá-Jacobo et al. (2019) [228]
Six regulated miRs identified, including hsa-miR-24-3p (↓) hsa-miR-34a-5p (↓)	Measured 14 miRNAs predicted to target diabetes susceptibility genes from peripheral blood	Study conducted in Greece	Controls (37) and T2D (40)	T2D patients compared to controls	Kokkinopoulou et al, (2019) [229]
Three regulated miRs identified, including hsa-miR-24-3p (↓)	Measured 84 T2D-related miRNAs in peripheral blood	Study conducted in Greece	Controls (37) and T2D (40)	T2D patients compared to controls	Avgeris et al. (2020) [230]
Eight regulated miRs identified: no matches to Table 1	Circulating microRNA sequenced from plasma samples	Prediabetic subjects selected from METabolic Syndrome In Men (METSIM) study. Subjects were matched between groups	Prediabetic patients that did not develop T2D (145). Prediabetic patients that developed T2D (145)	Prediabetic patients that developed T2D compared to prediabetic patients that did not develop T2D	Ghai et al. (2019) [231]
Three regulated miRs identified: no matches to Table 1	Measured 179 miRNAs from plasma samples	All subjects selected from Diabetes Prediction and Screening Observational Study (DIAPASON)	Control (9), impaired glucose tolerance (IGT) (9) and T2D (9)	T2D patients compared to controls	De Candia et al. (2017) [232]
Seven regulated miRs identified: no matches with Table 1	Measured 372 mature miRNAs from serum samples	Subjects recruited from First Affiliated Hospital of Guangzhou University of Chinese Medicine	Control (5) and T2D (10)	T2D patients compared to Controls	Yang et al. (2017) [233]
No match with Table 1	Measured let-7b-5p in serum samples	Subjects were recruited from the All-New Diabetics in harjah and Ajman (ANDISA) study	Control (25) and T2D without complications (29) and T2D patients with complications (27)	T2D patients without complications compared to controls	Aljaibeji et al. (2020) [234]
No match with Table 1	Expression of miR-20b and miR-17-3p were measured in serum samples	Subjects were selected from Internal Medicine and Opthalmology departments of Fayoum University, Fayoum, Egypt	Control (81), non-diabetic retinopathy (30), diabetic retinopathy (50)	Diabetic retinopathy patients compared to non-diabetic retinopathy patients	Shaker et al. (2019) [235]
No match with Table 1	MicroRNA-126 was measured in serum samples	Subjects recruited from Department of Endocrinology in the Hospital of Harbin Medical University, Harbin, China	Control (138), IGT (82), IFG (75) and T2D (60)	T2D patients compared to controls	Liu et al. (2014) [236]
36 regulated miRs identified, including hsa-miR-182-5p (↓)	Circulating miRNAs in plasma samples were sequenced	Participants selected from the Japanese American Community Diabetes Study (JACDS). Controls were matched to patients	Control (5), T2D patients (5)	T2D patients compared to controls	Wander et al. (2020) [237]
No match with Table 1	Measured miR-7 in serum samples	T2D patients that were newly diagnosed or during drug withdrawal. Controls were matched to patients	Controls (74) and T2D (152)	T2D patients compared to controls	Wan et al. (2017) [238]
Match with Table 1: hsa-miR-146a (↑)	Measured miR-146a and miR-155 in gingival crevicular fluid	Study conducted in Belgrade	Control (24), chronic periodontitis (24), peridontally healthy with T2D (24) and T2D patients with chronic periodontitis (24)	T2D patients compared to controls	Radović et al. (2018) [239]
Seven regulated miRs identified: no match with Table 1	Measured 372 miRNAs in serum samples	Subjects were selected from Vorarlberg Institute for Vascular Investigation and Treatment (VIVIT) cohort	Controls (43), IFG (43) and T2D (43)	T2D patients compared to controls	Jaeger et al. (2018) [240]
Match with Table 1: hsa-miR-146a (↓)	Measured miR-146a in PBMC and plasma samples	Study conducted in Tehran	Controls (30) and T2D (30)	T2D patients compared to controls	Alipoor et al. (2018) [241]
Two regulated miRs identified: no match with Table 1	Measured miRNA-9 and miRNA-370	Subjects were recruited from Al-Qasr AlEiny Teaching hospitals, Cairo, Egypt	Controls (50), T2D (50), CVD (50) and T2D with CVD (50)	T2D patients compared to controls	Motawae et al. (2015) [242]
hsa-miR-375 (↑)	Measured mature miRNA-375	Subjects were recruited from Departments of Endocrinology and Metabolism Shihezi University School of Medicine, China	Control (100) and T2D (100)	T2D patients compared to controls	Yin et al.(2017) [243]
34 regulated miRs identified, including hsa-miR-21-5p (↑) hsa-miR-146a-5p (↑)	All small RNA sequenced in corneas	Subject tissues were obtained from the National Disease Research Interchange (NDRI)	Control (10) and T2D (12)	T2D patients compared to controls	Kulkarni et al. (2017) [244]
Ten regulated miRs identified, including hsa-miR-146a (↓)	Circulating RNA sequenced from serum samples	Subjects recruited from the Diabetes Specialities Centre, Department of Endocrinology, Zhejiang Provincial Hospital or TCM, Hangzhou, China	Control (5) and T2D (5)	T2D patients compared to controls	Yang et al. (2014) [245]
Six regulated miRs identified: no matches with Table 1	Measured 24 miNAs selected through literature	Subjects selected from CORDIOPREV study	Control (355) and T2D (107)	T2D patients compared to Controls	Jiménez-Lucena, Rangel-Zúñiga et al. (2018) [246]
Four regulated miRs identified, including hsa-miR-145 (↓)	Measured miRNAs previously identified with a link to diet and developing T2D	Subjects selected from CORDIOPREV study comparing low-fat high-complex-carbohydrate (LFHCC) diet and Mediterranean diet	Control (355) and T2D (107)	T2D patients on LFHCC diet compared to controls on LFHCC diet	Jiménez-Lucena et al. (2021) [247]
No match with Table 1	Measured miR-103b in serum samples	Patients that had previously been assessed at the Department of Endocrinology, Alliliated Hospital of Southwest Medical University, Luzhou, Sichuan, China	Control (46), prediabetes (48), non-complicated T2D (43) and T2D and coronary heart disease (CHD) (36)	Non-complicated T2D patients compared to controls	Luo et al. (2015) [248]
No match with Table 1	Measured miR-15a in peripheral blood samples	Subjects recruited from King Abdullah University Medical Centre, Bahrain	Control (24), prediabetes (22) and T2D (24)	T2D patients compared to controls	Al-Kafaji et al. (2015) [249]
No match with Table 1	Exiqon qPCR panels used on serum samples	Study was conducted in Guangzhou, China	Control (25), T2D (25), obese (25) and T2D and obese (25)	T2D patients compared to controls	Wu et al. (2015) [250]
No match with Table 1	Measured miR-18a and miR-34c in PBMC samples	Subjects recruited from Beijing Xuanwu Hospital, Capital Medical University	Control (105), IFG (74) and T2D (117)	T2D patients compared to controls	Wang et al. (2017) [251]
One regulated miR detected: no match with Table 1	Measured 47 circulating miRNAs selected from pilot study in serum samples	Subjects were selected from the fasting cohort of the Malmö Diet and Cancer cardiovascular cohort (MDC-CC)	Control (259), CVD (169) and T2D (140)	T2D patients compared to Controls	Gallo et al. (2018) [252]
Regulated miR matches with Table 1: hsa-miR-375 (↑)	Measured miR-375 in serum samples	Subjects were selected from the First Affiliated Hospital of Shihezi University School of Medicine, Shihezi, Xinjiang, China	Controls (100) and T2D (100)	T2D patients compared to controls	Sun et al. (2014) [253]
No match to Table 1	Measured miR-155 in serum samples	Subjects were recruited from the Diabetes clinic of the Diabetes Research Centre, Endocrinology and Metabolism Research Institute, Tehran University of Medical Sciences	Controls (42) and T2D (45)	T2D patients compared to controls	Mahdavi et al. (2018) [254]
One regulated miR identified: no match to Table 1	Measured five miRNAs associated with T2D in serum samples	Subjects recruited from Outpatient Department if Laboratory Medicine, Chronic Disease Hospital of Nanshan District, Shenzhen, China	Control (3), prediabetes (30) and T2D (30)	T2D patients compared to controls	Zhang et al. (2013) [255]
Two regulated miRs identified, including hsa-miR-21-5p (↓)	Measured miR-21-5p and mR-126-3p in serum samples	All subjects reported a Mediterranean diet	Controls (107), T2D with no Complications (76) and T2D with complications (117)	T2D patients compared to controls	Olivieri et al. (2015) [256]
Nine regulated miRs identified, including hsa-miR-21 (↑) hsa-miR-24 (↑)	Measured miRNAs commonly found in plasma	Subjects selected from the impact of Migration and Ethnicity on Diabetes in Malmö (MEDIM) study	Control (119), T2D (33)	T2D patients compared to controls	Wang et al. (2014) [257]
No match with Table 1	Measured microvesicle miR-126 in serum samples	Subjects recruited from the Division of Metabolic Diseases of the University of Padua	Control (53), prediabetes (39) and T2D (68)	T2D patients compared to controls	Gianella et al. (2017) [258]
Four regulated miRs identified: no matches with Table 1	Four miRNAs selected following pilot study. Measured in plasma samples	Subjects recruited from People’s Hospital of Guizhou Privince, Guiyang, China	Control (50), prediabetes (50) and T2D (50)	T2D patients compared to controls	Yan et al. (2016) [259]
No match with Table 1	Measured miR-31, miR-93, miR0199a and miR-146a in plasma samples	Subjects recruited from the Geriatric Endocrinology Department of PLA General Hospital	Control (64), prediabetes (64) and T2D (64)	T2D patients compared to controls	Yan et al. (2014) [260]
Four regulated miRs detected including: hsa-miR-21 (↑) hsa-miR-24 (↑) hsa-miR-34a (↑)	Assessed a Diabetes-related human miRNA panel in serum samples	Subjects selected from the Outcome Reduction with Initial Glargine Intervention (ORIGIN) trial	Control (20), prediabetes (21) and T2D (17)	T2D patients compared to controls	Nunez Lopez et al. (2017) [261]
Two regulated miRs detected: no match with Table 1	Measured miR-144 and miR-223 in platelets and plasma samples	Subjects recruited at the Department of Endocrinology, Xiangya Hospital	Controls (30), T2D (56) and T2D with acute ischemic stroke	T2D patients compared to controls	Yang et al. (2016) [262]
No match with Table 1	Measured miRNAs that had been previously associated with T2D in plasma samples	Subjects recruited at the Outpatient Department of Laboratory Medicine, Chronic Disease Hospital of NanShan District, Shenzhen, China	Controls (20) and T2D (20)	T2D patients compared to controls	Zhang et al. (2015) [263]
Four regulated miRs identified: no match with Table 1	Serum miRNAs included in Exiqon Human panel 1	Subjects were recruited from Diabetes Specialities Centre, Chennai, India	Control (49), IGT (47) and T2D (49)	T2D patients compared to controls	Prabu et al. (2015) [264]
One regulated miR identified: hsa-miR-146a (↓)	Measured miR-155 and miR-146a in serum samples	Subjects were recruited in Quito, Ecuador	Control (40) and T2D (56)	T2D patients compared to controls	Baldeón et al. (2014) [265]
Match with Table 1: hsa-miR-146a (↓)	Measured miR-146a in PBMC samples	Subjects were recruited from Diabetes Specialities Centre, Chennai, India	Controls (35) and T2D (35)	T2D patients compared to controls	Lenin et al. (2015) [266]
No match with Table 1	Measured miR-126 in serum samples	Subjects recruited from Medical Biochemistry and Internal Medicine Departments of Zagazig University Hospitals, Egypt	Controls (100), IGT (86) and T2D (100)	T2D patients compared to controls	Rezk et al. (2016) [267]
Ten regulated miRs identified: no matches with Table 1	Assessed miRNAs in TaqMan Low-Density Arrays using serum samples	Subjects recruited in the Endocrinology Service of the Hospital Universitari Dr. Josep Trueta, Girona, Spain	Controls (45) and T2D (48)	T2D patients compared to controls	Ortega et al. (2014) [268]
Two regulated miRs detected: no matches with Table 1	Assessed miRNAs in circulating microparticles	Study conducted in Germany	Control (80) and T2D (55)	T2D patients compared to controls	Jansen et al. (2016) [269]
Four regulated miRs detected including: hsa-miR-21 (↑) hsa-miR-34a (↑)	Measured 28 pancreas-enriched miRNAs in plasma samples	Subjects were recruited from Florida, USA	Control (27), prediabetes (12), T2D (31), latent autoimmune diabetes in adults (LADA) (6) and T1D (16)	T2D patients compared to controls	Seyhan et al. (2016) [270]
Five regulated miRs identified: no matches with Table 1	Assessed miRNAs in TaqMan Low-Density Arrays using serum samples	Subjects were recruited from the Department of Endocrinology Jinling Hospital, Nanjing, China	Control (92) and T2D (184)	T2D patients compared to controls	Wang et al. (2016) [271]

## Data Availability

The data presented in thus study are openly available in Mendeley Data at doi:10.17632/jnz8h974gc.1.

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
