# Peer review of "MicroRNA Sequences Modulated by Beta Cell Lipid Metabolism: Implications for Type 2 Diabetes Mellitus"

_biology, 2021, doi:10.3390/biology10060534_

Round 1

Reviewer 1 Report

The review is generally well written, and suitably supported by appropriate figures and tables. There are  some areas that should be modified prior to publication however:

1) Use of the term "epigenetic landscape", line 20. This implies changes through DNA methylation or histone acetylation to influence gene expression, which is not the subject of this review. This would be better rewritten as "....understanding the landscape of gene expression which influences...." 

2) Line 87. The cited Ostenson et al manuscript details downregulation not only of SNAREs, but also of SNARE regulatory/accessory machinery and ion channels. These additional elements should also be stated and briefly discussed. 

3) Paragraph starting on line 216. In addition to statin and other drug modulators of cholesterol, glucotoxicity has also previously been shown to inhibit insulin secretion though downregulation of cholesterol synthesis and subsequent disruption of lipid rafts and associated secretory granule docking (Somanath et al, 2009, BBRC 389, 241-246). A statement to this effect should be added to the text. 

4) Line 223 and 224. Also line 433.  As per comment 1), it would be better to replace the term "epigenetic mechanisms" with "gene expression changes".

Author Response

Our thanks to the reviewer for taking the time to read and critique this paper.

  1. We accept the reviewer's reservations on this point and have made the alteration suggested (line 20).
  2. We have included details from the publication by Ostenson et al (2006), which now arise in lines 88-91.
  3. We have included the reference by Somanath et al (2009) in lieu of our own output, and discuss the findings in lines arising 226-234.
  4. We have altered the text, as per point 1, now arising at lines 235-236 and 449

Reviewer 2 Report

This extensive and interesting review addresses a topic of great interest. The bibliography is well selected and the information is well presented . Figures and tables help to understand the information. In my opinion it is a good review ready to be published. 

Author Response

We thank the reviewer for their kind comments and for taking time to review our paper.

Reviewer 3 Report

This is an excellent review of B cell lipid metabolism and Micro RNA.. Although the concept that diabetes is first and foremost a disease of lipid metabolism was raised many years ago this review  records the massive work now being undertaken to understand the dyslipidaemia of diabetes and its consequence on the B Cell, There are excellent figs and tables and a bibliography that is extensive and so impressive. This article will be of huge benefit to those undertaking research into the metabolism of the B Cell and for those wanting an update on Micro RNAs.

Author Response

We thank the reviewer for their kind comments, and for taking the time to review our paper

Reviewer 4 Report

The review is well written and is very useful for researchers in this field to have an overview of the target miRNA sequences to develop RNA-based therapeutics for T2 diabetes. Well done to the authors. 

opens up the possibilities

Author Response

(The authors gave the same response as above.)
